# Decrease in Ambient Fine Particulate Matter during COVID-19 Crisis and Corresponding Health Benefits in Seoul, Korea

**DOI:** 10.3390/ijerph17155279

**Published:** 2020-07-22

**Authors:** Changwoo Han, Yun-Chul Hong

**Affiliations:** 1Department of Preventive Medicine, Chungnam National University College of Medicine, Daejeon 35015, Korea; cwohan@cnu.ac.kr; 2Department of Preventive Medicine, Seoul National University College of Medicine, Seoul 03080, Korea; 3Institute of Environmental Medicine, Seoul National University Medical Research Center, Seoul 03080, Korea; 4Environmental Health Center, Seoul National University College of Medicine, Seoul 03080, Korea

**Keywords:** COVID-19, particulate matter, lockdown, health burden, mortality, Korea

## Abstract

Both domestic emissions and transported pollutants from neighboring countries affect the ambient fine particulate matter (PM_2.5_) concentration of Seoul, Korea. Diverse measures to control the coronavirus disease 2019 (COVID-19), such as social distancing and increased telecommuting in Korea and the stringent lockdown measures of China, may reduce domestic emissions and levels of transported pollutants, respectively. In addition, wearing a particulate-filtering respirator may have decreased the absolute PM_2.5_ exposure level for individuals. Therefore, this study estimated the acute health benefits of PM_2.5_ reduction and changes in public behavior during the COVID-19 crisis in Seoul, Korea. To calculate the mortality burden attributable to PM_2.5_, we obtained residents’ registration data, mortality data, and air pollution monitoring data for Seoul from publicly available databases. Relative risks were derived from previous time-series studies. We used the attributable fraction to estimate the number of excessive deaths attributable to acute PM_2.5_ exposure during January to April, yearly, from 2016 to 2020, and the number of mortalities avoided from PM_2.5_ reduction and respirator use observed in 2020. The average PM_2.5_ concentration from January to April in 2020 (25.6 μg/m^3^) was the lowest in the last 5 years. At least −4.1 μg/m^3^ (95% CI: −7.2, −0.9) change in ambient PM_2.5_ in Seoul was observed in 2020 compared to the previous 4 years. Overall, 37.6 (95% CI: 32.6, 42.5) non-accidental; 7.0 (95% CI: 5.7, 8.4) cardiovascular; and 4.7 (95% CI: 3.4, 6.1) respiratory mortalities were avoided due to PM_2.5_ reduction in 2020. By considering the effects of particulate respirator, decreases of 102.5 (95% CI: 89.0, 115.9) non-accidental; 19.1 (95% CI: 15.6, 22.9) cardiovascular; and 12.9 (95% CI: 9.2, 16.5) respiratory mortalities were estimated. We estimated that 37 lives were saved due to the PM_2.5_ reduction related to COVID-19 in Seoul, Korea. The health benefit may be greater due to the popular use of particulate-filtering respirators during the COVID-19 crisis. Future studies with daily mortality data are needed to verify our study estimates.

## 1. Introduction

The world is facing one of its gravest challenges due to the coronavirus disease 2019 (COVID-19). After the first unknown pneumonia cases detected in Wuhan, China, in December 2019, scientists identified new species of the zoonotic coronavirus, severe acute respiratory syndrome coronavirus 2 (SARS-COV-2), causing COVID-19 [1,2]. COVID-19 may lead to a fatal respiratory disease in the elderly and in persons with preexisting chronic diseases, but the symptoms can be mild in young and healthy individuals [3]. According to the report from the Chinese Center for Disease Control using the 44,672 confirmed COVID-19 cases, the case fatality rate was 2.3%, lower compared to other coronaviruses causing severe acute respiratory syndrome (SARS, 9.6%) and Middle East respiratory syndrome (MERS, 35.5%) [4,5]. However, SARS-COV-2 is highly infectious, with an estimated reproduction number ($R_{0})$ over 3, causing more mortalities than any other coronaviruses [5,6,7]. Due to the wide geographical spread of the virus and the growing international concern, the World Health Organization (WHO) declared COVID-19 outbreak as a pandemic on 12 April 2020. As of 24 April 2020, COVID-19 has killed more than 190,800 people, with 4.6 million confirmed cases worldwide. Billions of people were instructed or forced to stay at home while countries locked their borders to varying degrees to control COVID-19.

After the first incidence of COVID-19 on 20 January 2020, Korea has controlled the outbreak relatively well compared to other countries. Although there was a surge of COVID-19 in Daegu City, which peaked at over 900 incident cases on 29 February, the daily increase in new cases in Korea had dropped below 10 by the end of April [8]. Without imposing extreme measures to restrict the movement or freedom of the citizens, Korea was able to flatten the curve of the new cases. This was achieved by a massive number of tests conducted for early detection, by applying rigorous epidemiological investigation, and sharing information using the state of the art information and communication technologies [8,9]. These all occurred with the voluntary participation of social distancing and personal hygiene practices (i.e., wearing respirators in public space and washing hands) by the citizens. To address the shortage and ensure the equal distribution of respirators, the Korean government adopted a “5-day rotation system for a respirator” to provide opportunities for people to purchase a respirator at pharmacies on designated days according to their birth years.

Recently, the diverse news media, journal papers, and reports posted on the preprint server medRxic have indicated that the stringent lockdown measures not only helped in controlling the spread of this highly infectious virus, but also helped to decrease the ambient air pollution levels, including particulate matter less than 2.5 μm in diameter (PM_2.5_) and nitrogen dioxide (NO_2_) [10,11]. With the lockdown of industrial activities and transportation in China for about 35 days, there was a 30.1 and 9.2 μg/m^3^ decrease in PM_2.5_ in Wuhan and Beijing, respectively [12]. One report estimated a 12.9 and 18.9 μg/m^3^ decrease in NO_2_ and PM_2.5_ in China after the massive population quarantine [13]. Satellites of National Aeronautics and Space Administration (NASA) and European Space Agency detected a 10 to 20% decrease in NO_2_ levels over eastern and central China after the quarantine [14]. The Movement Control Order of the Malaysian government to encourage their citizens to work at home and suspend industrial activities caused a decrease in PM_2.5_ levels [15]. Lockdown measures such as school closure, work at home, avoiding gatherings, shutting down commerce, and limiting public transportation in the city of Rio de Janeiro and São Paulo State caused a reduction in carbon monoxide and NO_2_ levels [16,17]. Another report estimated a 9%, 29%, and 11% global reduction in ambient PM_2.5_, NO_2_, and ozone levels, respectively, after the COVID-19 crisis [18].

A recent announcement by Korea’s Ministry of Environment revealed a reduction in the average PM_2.5_ in Korea from December 2019 to April 2020 by 27% (9 μg/m^3^) compared to that of the previous year [19]. Although the report highlighted the effectiveness of “the seasonal particle pollution measures”, which was newly implemented in December 2019, the marked PM_2.5_ decrease observed in China as well as around the world suggest that the measures to control COVID-19 may be the largest contributing factor behind the improved air quality. In addition, a recent modeling study showed that around 70% of PM_2.5_ concentration of Seoul is affected by the emission from China during the days with severe PM_2.5_ concentration in spring [20].

Therefore, we initiated this study with a reasonable assumption that social distancing and increased telecommuting in Korea, in addition to the stringent lockdown measures and reduced industrial activities in China, may have reduced the PM_2.5_ levels in Seoul, the capital city of Korea. Furthermore, the respirators supplied and distributed by the Korean government were effective in filtering PM_2.5_ at least 80% of the 0.6 μm nonoil particulates. Therefore, each person would be exposed to a lower level of PM_2.5_ compared to the period before the COVID-19 crisis by using the particulate filtering respirator. Based on this assumption, we estimated the acute health benefits of PM_2.5_ reduction and changes in public behavior (wearing of a respirator) during the COVID-19 crisis using the health impact assessment methodology. The aim of this study was to use currently available data to estimate the acute health benefits of PM_2.5_ reduction and changes in public behavior, which were changes experienced by Korean citizens in their daily lives during the COVID-19 crisis.

## 2. Materials and Methods

### 2.1. Study Design

The health impact assessment requires data on exposure, mortality, and the population. Although Korea shares the real-time air pollution monitoring data with the public, the mortality data are not shared simultaneously. Therefore, we estimated the mortality rates over the last 2 years (2019 and 2020) based on the mortality rates of earlier years. With the PM_2.5_ monitoring data, population data, and estimated mortality rates in Seoul, we estimated the health benefits based on the PM_2.5_ reduction levels in 2020. By using previous respirator intervention study results [21], we estimated the health benefits of decreased PM_2.5_ and respirator use in 2020.

To estimate the PM_2.5_ decrease in year 2020, we first calculated the average PM_2.5_ concentration during the first 4 months of 2020 and compared it with the concentration in the same months, each year, from 2016 to 2019. By using the attributable fraction (AF) method [22], we estimated the mortality burden attributable to the acute ambient PM_2.5_ exposure in the first 4 months of each year, and the number of mortalities avoided due to the observed PM_2.5_ reduction in 2020.

To calculate the mortality burden attributable to PM_2.5_, we obtained the population, mortality, and air pollution monitoring data of Seoul from publicly available databases. The relative risks (RR) derived from previous time-series studies, which evaluated the association between ambient PM_2.5_ exposure and cause-specific mortalities were reviewed to retrieve the beta estimates of the concentration–response functions [23,24]. To consider the effects of public respirator use, we referred to previous intervention study, which evaluated the difference between individual PM_2.5_ exposure level by the particulate respirator use [21].

Several important assumptions were made for our study. Because mortality data were unavailable for the last two years (2019 and 2020), we assumed that mortality rates in Seoul from January to April had not changed in the last 5 years. In Korea, the daily mortality data for a typical year become publicly available at least 1.5 years after the year ends. Therefore, we estimated the mortality rates of Seoul from January to April in 2019 and 2020 by averaging the mortality rates of the same months from 2016 to 2018. Because the mortality rates are closely related to the structure of the population, we also assumed that the composition of the population by age groups had not changed during our study period.

Second, we assumed that the effects of PM_2.5_ were limited to a single day, ignoring the delayed effect of PM_2.5_. PM_2.5_ is known to have lag effects of several days after the exposure [25]. However, because the daily mortality counts were unavailable for the years 2019 and 2020, it was impossible to estimate the daily levels accounting for lag effects of PM_2.5_. Therefore, we regarded the 4-month period as a whole and calculated the mortality burden for each year using the 4-month averages of PM_2.5_ concentrations and the estimated mortality rates of Seoul. Assumption focusing on a single day effect of PM_2.5_ may underestimate the overall mortality burden attributable to PM_2.5_. However, the avoided number of mortalities due to PM_2.5_ reduction during the COVID-19 crisis may not be biased because we focused on the changes in PM_2.5_ levels with the assumption of a linear concentration–response function.

Third, we assumed that the effects of PM_2.5_ concentration on health outcomes did not change during the study period, despite the changes in personal behaviors (i.e., social distancing and decreased outdoor activities) due to the COVID-19 crisis. We adopted the RRs from the previous time-series studies conducted both in Seoul and that in 652 cities around the globe [23,24]. However, despite assuming that the slope of concentration–response function between PM_2.5_ and mortality remained unchanged, we assumed that wearing a particulate filtering respirator would decrease the absolute level of PM_2.5_ exposure of an individual.

This study was exempt from review by the Institutional Review Board of the Seoul National University Hospital, Korea (IRB No.: 2006-122-1133), because data used in our study were de-identified and publicly available.

### 2.2. Population and Mortality Data

The yearly residents’ registration data for January during the study period (2016 to 2020) and the cause-specific mortality data for January to April (2016 to 2018) were acquired from the publicly available databases, the Korean Statistical Information Service website, and the Korean Statistical Information Service MicroData Integrated Service website [26,27].

We used the International Classification of Disease, 10th revision codes to define the following cause-specific mortalities: non-accidental and specific disease mortality (A00-R00), cardiovascular disease mortality (I00-I99), and respiratory disease mortality (J00-J99). The number of deaths due to these disease categories during the 4 months (January to April) of each year was calculated and divided by the number of registered residents in January of the corresponding year, to calculate the 4-month average in mortality rates of Seoul.

### 2.3. PM_2.5_ and Meteorological Data

Ambient PM_2.5_ data of Seoul from January 2016 to April 2020 were accessed through the Airkorea website, which provides real-time air pollution monitoring data of Korea [28]. Korea operates 25 air pollution monitoring stations in Seoul, covering 25 basic administration districts. We acquired the hourly PM_2.5_ monitoring results from each station, and calculated Seoul’s daily and 4-month average of PM_2.5_ exposure levels for January to April each year from 2016 to 2020. Daily meteorological data such as ambient temperature, relative humidity, and wind speed from January 2016 to April 2020 were accessed through the National Climate Data Center website. The map of Seoul and the locations of the ambient PM_2.5_ as well as the weather monitoring stations are presented in Appendix A.

### 2.4. Concentration-Response Functions and Effects of Particulate Matter Filtrating Respirator

The beta estimates of the concentration—response functions were retrieved from the RRs of the previous time-series studies that evaluated the association between ambient PM_2.5_ exposure and cause-specific mortalities (Appendix A). We selected two recently published studies—one analyzing the data from cities around the globe, and the other limited to Seoul. In brief, the Multi-City Multi-Country (MCC) collaborative research network gathered the daily mortality rates and PM_2.5_ data from 499 cities in 16 countries. The researchers found that a 10 μg/m^3^ increase in a 2-day moving average of ambient PM_2.5_ was associated with 0.68% (95% confidence interval (CI): 0.59, 0.77); 0.55% (95% CI: 0.45, 0.66); and 0.75% (95% CI: 0.53, 0.95) increase in the daily non-accidental, cardiovascular, and respiratory mortalities, respectively [23].

The study conducted in Seoul used the daily PM_2.5_ concentrations and mortality data of the city from 2006 to 2012. A 10 μg/m^3^ increase in ambient PM_2.5_ was associated with an increase in non-accidental mortality (0.33% (95% CI: 0.01, 0.66)), cardiovascular mortality (0.76% (95% CI: 0.12, 1.41)), and respiratory mortality (1.77% (95% CI: 0.55, 3.01)) on the same day in Seoul [24].

In a previous intervention study with 21 female elderlies in Korea, we evaluated the effects of a particulate-filtering respirator on cardiopulmonary function by using the crossover study design [21]. The subjects were instructed to use (intervention period) or not use the respirator (control period) for six consecutive days and had a medical examination on the last day of each period. By using the disposable particulate respirators (capable of filtering 80% of the 0.6 μm nonoil particulates), we found that the average level of personal exposure to PM_2.5_ had decreased by 27.4% (9.0 μg/m^3^ reduction) during the respirator use, and even the outdoor (27.4–28.8 μg/m^3^) and 24-h personally monitored PM_2.5_ levels (18.7–20.1 μg/m^3^) were similar between intervention and control periods. By referring to this value, we estimated that the effects of respirator use during COVID-19 crisis decreased the personal PM_2.5_ exposure level by 27.4% in addition to the decrease in ambient PM_2.5_ levels observed during 2020.

### 2.5. Statistical Analysis

With the estimated number of mortalities and monitored PM_2.5_ concentrations from January to April each year, we used the AF method to estimate the mortality burden attributable to ambient PM_2.5_ levels [22].
(1)$$Excess\;deaths\;by\;PM_{2.5}=\left(1-exp^{-\beta \times \Delta C}\right)\;\times \;Number\;of\;deaths$$

β is the coefficient derived from the RRs in the previous time-series studies and ΔC refers to the changes in the PM_2.5_ concentrations under different counterfactual scenarios. The number of deaths from January to April for the years 2016 to 2018 was obtained from the mortality database, while those of 2019 and 2020 were calculated by multiplying the number of registered residents with the mortality rates estimated based on the mortality rates of 2016 to 2018.

To calculate the number of excess deaths attributable to the acute PM_2.5_ exposure from January to April each year from 2016 to 2020, we defined 2.4 μg/m^3^ as the concentration with the minimum health risk, which is the theoretical minimum risk exposure level, indicating no health benefits for reducing PM_2.5_ below the level based on prior epidemiological studies [29]. Therefore, ΔC in the equation indicates the difference between average PM_2.5_ concentrations monitored each year from January to April and 2.4 μg/m^3^. On the other hand, to calculate the avoided mortality due to PM_2.5_ reduction in 2020, we defined ΔC as the estimated reduction in PM_2.5_ from January to April in 2020 compared to the same months in 2016–2019.

To calculate the reduction in PM_2.5_ in 2020, we used the linear regression models assuming the normal distribution of PM_2.5_ levels. In model 1, the amount of reduction was estimated based on a simple comparison of the average value for 2020 with the average for the years 2016–2019. In model 2, we adjusted for meteorological variables (daily average temperature, relative humidity, and wind speed), and in model 3, we adjusted for the meteorological variables, years (as a continuous variable), and the months (as a categorical variable). In model 4, we adjusted for the meteorological variables and assumed that the average level of personal exposure to PM_2.5_ had decreased by 27.4% in 2020, to account for the widespread use of particulate filtrating respirator.

All analyses were conducted with SAS version 9.4 (SAS Institute Inc., Cary, NC, USA), and figures were drawn using the R statistical software (Version 3.6.1; R Foundation for Statistical Computing, Vienna, Austria). The level of statistical significance was set at a *p*-value of less than 0.05.

## 3. Results

Table 1 and Figure 1 show the 2016 to 2020 Seoul data for the daily PM_2.5_ concentrations, ambient temperature, relative humidity, wind speed, and the number of days that the PM_2.5_ concentration was above the WHO and Korea 24-h average standards. The average ambient temperature from January to April in 2020 was the highest compared to those of the previous 4 years, while the relative humidity and wind speed of 2020 were similar to that of 2016 and 2017.

The average PM_2.5_ concentration from January to April in 2020 (25.6 μg/m^3^) was the lowest in the last 5 years. Overall, the PM_2.5_ concentrations above the WHO (25 μg/m^3^) and the Korean standards (35 μg/m^3^) in 2020 were for 55 days (45.5%) and 25 days (20.7%) respectively, which were the least number of days in the past 5 years. Figure 1 shows the dramatic decrease in the daily PM_2.5_ concentrations as well as the number of days with spiking PM_2.5_ concentrations in 2020 compared to the previous 4 years.

Table 2 shows the number of registered residents, estimated number of deaths, and mortality rates used in the study. The average number of registered residents in Seoul in January each year was 9,860,115. The average non-accidental, cardiovascular, and respiratory mortalities per 100,000 persons in Seoul were 139.2, 32.0, and 16.1 from January to April in 2016 to 2018, respectively. We used these mortality rates to estimate the number of deaths for 2019 and 2020 assuming that these rates remained unchanged. We estimated that 13,549; 3115; and 1567 persons died in Seoul from January to April in 2020 due to non-accidental, cardiovascular, and respiratory diseases, respectively.

Figure 2, Appendix A, and Appendix A show the number of mortalities attributable to PM_2.5_ exposure in Seoul from January to April each year from 2016 to 2020. By using the MCC study’s RRs, the daily exposure to PM_2.5_ in 2020 caused 211.4 (95% CI: 183.7, 239.0); 39.4 (95% CI: 32.3, 47.2); and 26.6 (95% CI: 19.1, 34.0) deaths due to non-accidental, cardiovascular, and respiratory diseases, respectively (Appendix A). The mortality attributable to the daily PM_2.5_ exposure was the lowest in 2020 compared to those of the previous 4 years (Figure 2). The results using RRs from the Seoul study are summarized in Appendix A and Appendix A.

Table 3 shows the estimated PM_2.5_ reduction levels and the avoided mortality due to the PM_2.5_ exposure in 2020 compared to those from 2016–2019. By simply comparing the average values, a −5.6 μg/m^3^ (95% CI: −9.0, −2.3) change in ambient PM_2.5_ was observed in Seoul from January to April in 2020 compared to the same months in the previous 4 years (model 1). By adjusting for meteorological variables, a −4.1 μg/m^3^ (95% CI: −7.2, −0.9) change in ambient PM_2.5_ was estimated (model 2). By further adjusting for years and months, a −15.1 μg/m^3^ (95% CI: −27.1, −3.2) change in ambient PM_2.5_ was estimated (model 3). With the conservative estimation of a 4.1 μg/m^3^ decrease in PM_2.5_ and RRs from the MCC study, we found that 37.6 (95% CI: 32.6, 42.5) non-accidental; 7.0 (95% CI: 5.7, 8.4) cardiovascular; and 4.7 (95% CI: 3.4, 6.1) respiratory mortalities were avoided because of the reduction in PM_2.5_ from January to April in 2020 compared to those of the previous 4 years.

By considering the effect of the meteorological variables and the particulate respirator use during 2020 (model 4), we estimated that the personal exposure to ambient PM_2.5_ decreased by −11.2 μg/m^3^ (95% CI: −14.3, −8.2) from January to April in 2020 compared to the previous 4 years. The corresponding health benefits were decreases of 102.5 (95% CI: 89.0, 115.9) non-accidental; 19.1 (95% CI: 15.6, 22.9) cardiovascular; and 12.9 (95% CI: 9.2, 16.5) respiratory mortalities.

## 4. Discussion

We observed at least a 4.1 μg/m^3^ decrease in ambient PM_2.5_ concentration in Seoul from January to April in 2020 compared to the same months in 2016–2019. We estimated that 37 persons were saved due to the reduction in PM_2.5_ during the 4-month period. Because using a particulate-filtrating respirator may decrease the absolute level of PM_2.5_ exposure for an individual, the health benefit related to air pollution during the COVID-19 crisis may be larger than our current estimation of 37 persons.

There are several possible explanations for the decrease in PM_2.5_ in Seoul. First, public behavioral changes such as social distancing and reduced outdoor activities to limit COVID-19 transmission may have decreased the air pollution levels. According to the mobility data based on the map navigation application on smartphones, both walking and driving by the public were decreased in Seoul after the COVID-19 crisis (Appendix A). The daily amount of traffics entering the highways and the number of citizens using the Seoul Metropolitan Area subway from January to April in 2020 dropped by 6.1% and 28.4%, respectively, compared to the same months in the period 2016–2019 (Appendix A). In addition, industrial activities such as the number of operating factories may have decreased due to the limited consumer demand during COVID-19 crisis, which may have resulted in decreased domestic emissions [30].

Similar improvements in air quality were observed around the world since the COVID-19 crisis, with an estimated reduction of 9%, 29%, and 11% of ambient PM_2.5_, NO_2_, and ozone levels, respectively [18]. The reductions in carbon monoxide and NO_2_ levels were observed after the partial lockdown (school closure, work from home, avoiding gatherings, shutting down commerce, and limiting public transportation) in the city of Rio de Janeiro and São Paulo State [16,17]. During the Movement Control Order (work from home and suspend industrial activities) to isolate the source of COVID-19 in Malaysia, up to a 58.4% decrease in PM_2.5_ was observed [15]. By analyzing air quality monitoring data of 22 cities in India, 43% and 18% decreases in PM_2.5_ and NO_2_ were observed during the COVID-19 crisis [31].

The air pollution levels in Seoul cannot be evaluated without considering the effects of the neighboring countries, China and North Korea. By evaluating the source contribution of PM_2.5_ on days with severe PM_2.5_ concentrations (with 24-h average PM_2.5_ concentration of over 100 μg/m^3^) in Seoul, China contributed to the PM_2.5_ concentrations by up to 70% while the domestic contribution was 21% [20]. In addition, around 15% of PM_2.5_ concentration in Seoul is affected by the emission from North Korea [32]. Among the 1638 mortalities attributable to the acute exposure to high levels of PM_2.5_ in Korea in 2016, at least 258 and 26 deaths were estimated to have been due to the emissions from China and North Korea, respectively [33].

Because the air quality in China improved dramatically during the COVID-19 quarantine (10 February to 14 April 2020) [12,13,14,34], and considering the fact that China’s contribution to Seoul’s PM_2.5_ concentration is generally greater in the spring and winter [33], the decrease in PM_2.5_ observed in Seoul from January to April 2020 may partially be explained by the effects of China’s rigorous quarantine measures and decreased industrial activities during the COVID-19 crisis. If the observed decrease in PM_2.5_ levels in Seoul in 2020 is indeed due to the changes related to domestic responses as well as China’s response against COVID-19, the estimated mortality benefit from the lowered PM_2.5_ levels (37 persons) outweighs the number of the direct casualties from COVID-19 in Seoul (2 persons till 30 April 2020). Similar paradoxical phenomena, and a massive decrease in air-pollution-related mortalities and morbidities during the COVID-19 crisis are expected worldwide [14].

Another plausible explanation for the decrease in PM_2.5_ concentration in Seoul in 2020 is the governmental effort to reduce the domestic sources of PM_2.5_ [19]. With consultations with relevant ministries, a comprehensive set of measures to control the particulate matter in 2020 to 2024 was finalized on 1 November 2019 [35]. One of the measures is the seasonal management of the domestic sources of PM_2.5_ from December to April, when PM_2.5_ levels are usually high [36]. Efforts to reduce the domestic emission of PM_2.5_ include the shutting down of the coal-fired power plants, voluntary reduction in emissions from business sites, nationwide surveillance of emission sources, designation of low-sailing zones, and transition to low sulfur fuels for ocean vessels. In addition to these comprehensive measures, an increase in rainfall and the number of days with high wind velocity may also help to explain the decrease in PM_2.5_ observed in Seoul from January to March 2020 [19]. However, the relative humidity and wind speed in 2020 were similar to those of 2016 and 2017; limiting the effects of meteorological factors on PM_2.5_ reduction.

We may not be able to distinguish the effects of diverse governmental measures from those of domestic and international changes related to COVID-19 on the reduced PM_2.5_ levels in Seoul. However, it is reasonable to assume that the domestic measures (social distancing and avoidance of outside activities in Korea) and international effects (decreased PM_2.5_ levels during the quarantine in China) related to COVID-19 may have played a significant role at the same time. We may be able to confirm the effects of COVID-19 by observing the air pollution levels after the COVID-19 measures are lifted and by evaluating whether the effects (decrease in air pollution) cease after the treatment (measures against COVID-19) ends [37].

Although not formally published as a journal article, several reports posted on medRxiv are showing the health benefits of reduced PM_2.5_ levels after the COVID-19 crisis. One report estimated that the PM_2.5_ and NO_2_ levels dropped by 18.9 and 12.9 μg/m^3^ in China during the COVID-19 quarantine period, which led to a decrease of 3214 PM_2.5_-related deaths and a decrease of 8911 NO_2_-related deaths [13]. By analyzing the air pollution data from over 10,000 monitoring stations around the globe, a 9%, 29%, and 11% reduction in ambient PM_2.5_, NO_2_, and ozone levels was estimated during February to April 2020, just after the global lockdown in response to COVID-19 [18]. The corresponding health benefits related to the decrease in air pollution levels were 7400 deaths and 6600 pediatric asthma cases.

After the first confirmed case of COVID-19 in Korea on 20 January 2020, the panic buying of particulate filtrating respirator led to instability in supply and demand. To tackle this issue, the Korean government adopted a “5-day rotation system for respirator” to provide equal opportunity for individuals to purchase two particulate-filtering respirators (particulate respirators capable of filtering at least 80% of the 0.6 μm nonoil particulates) per week. As the Korean government instructed its citizens to wear a respirator outside the house to control COVID-19, personal level of exposure to ambient PM_2.5_ may have decreased from wearing a particulate respirator. With the results of the previous intervention study, we estimated that the effects of respirator use during COVID-19 crisis decreased the personal PM_2.5_ exposure level by 27.4% in addition to the decrease in ambient PM_2.5_ levels. We estimated that this decrease led to 102 averted deaths related to PM_2.5_ during the 4-month period; this is higher compared to the conservative estimation of 37 lives saved with a 4.1 μg/m^3^ decrease in ambient PM_2.5_ concentration during the COVID-19 crisis estimated in our study. We believe that using a particulate filtrating respirator not only help to block the transmission of COVID-19, but also helped to limit the adverse health effects of PM_2.5_.

We have previously estimated that around 12,000 premature deaths were attributable to chronic PM_2.5_ exposure in Korea in 2015, when the annual average of PM_2.5_ concentration was 24.4 μg/m^3^ [38]. We have also estimated that 1763 deaths solely occurred in Seoul. However, our study is different from the previous study in terms of the fact that it addressed the acute effects of PM_2.5_ exposure by using the RRs from previous time-series studies. With the yearly mortality information and PM_2.5_ measurement data for the entire year, we may be able to estimate the chronic PM_2.5_ exposure burden for 2020 and compare the difference with previous years. If reduced PM_2.5_ levels are maintained throughout 2020, we may be able to see a marked decrease in PM_2.5_-related burden.

Several limitations should be noted for this study. First, we adopted the RRs from previous time-series studies and ignored the possibility that the association between PM_2.5_ and health outcomes may have changed during the COVID-19 crisis. The Korean government instructed its citizens to avoid outdoor activities or crowded areas, and to use a respirator outside the house. Due to social distancing and the use of a respirator, the beta coefficient of exposure–response relationship may have changed. If the daily mortality data become available in the future, we may be able to confirm the changes in beta coefficient by comparing the estimates from the time-series analyses before and during the COVID-19 crisis. In addition, although we tried to adjust for the effects of respirator by using the results from previous intervention study, we also assumed that the entire public were using the particulate respirator during the COVID-19 crisis, which is unlikely. Second, we ignored the daily lag effect of PM_2.5_. Because mortality data are not disclosed simultaneously, we were unable to conduct a day-to-day analysis accounting for the lag effects of PM_2.5_. With the full mortality data of 2020, a more precise estimation can be conducted by using the daily data rather than using the 4-month (January to April) data as a whole.

Due to insufficient data and assumptions used in our study, our study estimates may be biased. Assumptions regarding mortality rates and concentration-response functions have to be validated with a daily number of mortality data and PM_2.5_ monitoring data in the future. However, based on the currently available data, our study may offer a glimpse into the acute health benefits of PM_2.5_ reduction and changes in public behaviors, which are the tangible changes experienced by the citizens in their daily lives during the COVID-19 crisis.

## 5. Conclusions

We observed at least 4.1 μg/m^3^ decrease in ambient PM_2.5_ in Seoul from January to April 2020, and this decrease is believed to be the results of the changes related to COVID-19 crisis. With our conservative estimation, a total of 37 lives were saved due to the PM_2.5_ reduction in Seoul from January to April 2020 compared to the same period in previous years. However, the health benefits related to the decrease in PM_2.5_ may be greater because of the popular use of the particulate respirator by the public during the COVID-19 crisis in Korea. We may need to verify our study findings by observing the PM_2.5_ levels after the COVID-19 crisis and conducting studies with a full set of daily mortality data.

## Figures and Tables

**Figure 1 ijerph-17-05279-f001:**
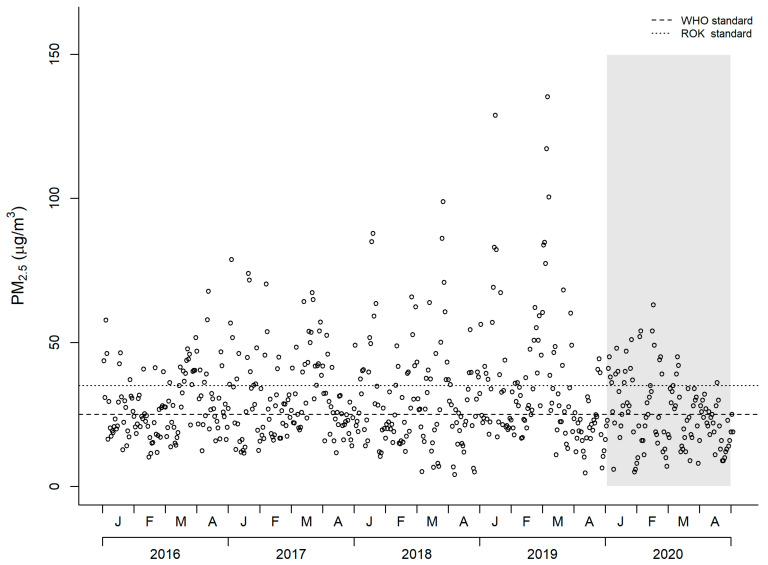
PM_2.5_ concentration in Seoul from year 2016 to 2020 (January to April).

**Figure 2 ijerph-17-05279-f002:**
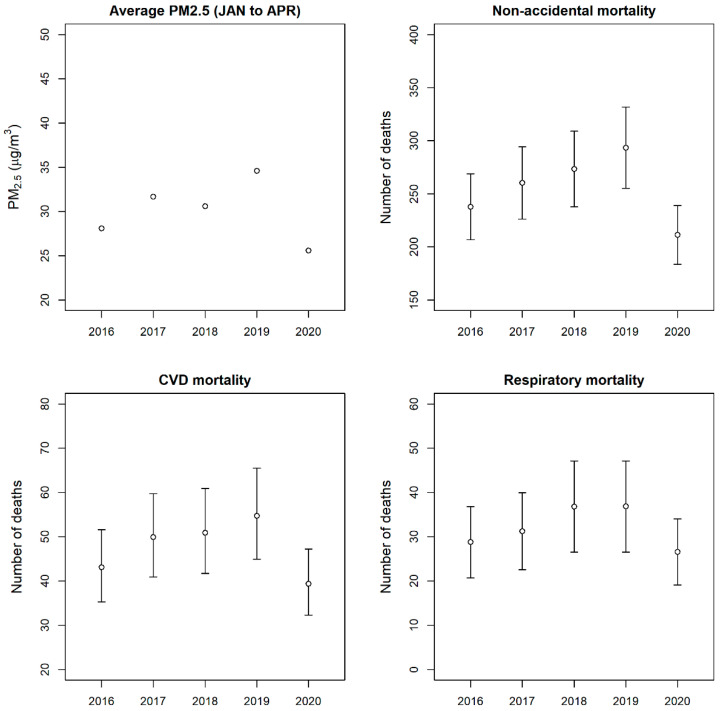
Average PM_2.5_ concentration of Seoul from January to April of 2016 to 2020 and estimated number of mortalities attributable to PM_2.5_ exposure (RRs from the MCC study were used for the estimation).

**Table 1 ijerph-17-05279-t001:** Average daily particulate matter (PM_2.5_) concentrations, temperature, relative humidity, wind speed, and number of days over the WHO (25 μg/m^3^) and Republic of Korea (ROK, 35 μg/m^3^) PM_2.5_ 24-h average standards in Seoul (January to April each year from 2016 to 2020).

Year	Days (N)	PM_2.5_ (μg/m^3^) ^(a)^	Temperature (°C) ^(a)^	Relative Humidity (%) ^(a)^	Wind Speed (m/s) ^(a)^	Days over WHO Standard ^(b)^	Days over ROK Standard ^(b)^
2016	121	28.1 (11.1)	4.5 (7.8)	52.7 (14.4)	2.5 (0.7)	64 (52.9)	33 (27.3)
2017	120	31.7 (15.2)	4.6 (7.1)	52.1 (12.7)	2.4 (0.7)	71 (59.2)	40 (33.3)
2018	120	30.6 (18.3)	3.9 (8.4)	51.8 (14.4)	2.0 (0.7)	64 (53.3)	40 (33.3)
2019	120	34.6 (23.6)	4.9 (6.0)	48.8 (14.6)	1.9 (0.6)	63 (52.5)	39 (32.5)
2020	121	25.6 (12.2)	5.8 (5.3)	52.4 (13.5)	2.5 (0.8)	55 (45.5)	25 (20.7)

^(a)^ Mean and standard deviation are presented, ^(b)^ Number of days and percentage are presented.

**Table 2 ijerph-17-05279-t002:** Number of registered population, number of deaths, and mortality rates used in this study.

Year	2016	2017	2018	2019	2020
Registered population at January, Seoul	10,018,537	9,930,478	9,851,767	9,766,288	9,733,509
Non-accidental mortality (A00-R00, January to April)					
Number of deaths	13,776	13,243	14,445	13,595 ^(a)^	13,549 ^(a)^
Mortality rate (per 100,000)	137.5	133.4	146.6	139.2 ^(b)^	139.2 ^(b)^
Cardiovascular disease mortality (I00-I99, January to April)					
Number of deaths	3080	3129	3316	3125 ^(a)^	3115 ^(a)^
Mortality rate (per 100,000)	30.7	31.5	33.7	32.0 ^(b)^	32.0 ^(b)^
Respiratory disease mortality (J00-J99, January to April)					
Number of deaths	1533	1462	1789	1572 ^(a)^	1567 ^(a)^
Mortality rate (per 100,000)	15.3	14.7	18.2	16.1 ^(b)^	16.1 ^(b)^

^(a)^ Calculated based on the estimated mortality rate, ^(b)^ Estimated by averaging year 2016–2018 mortality rate.

**Table 3 ijerph-17-05279-t003:** Estimated PM_2.5_ reduction levels and avoided mortality due to PM_2.5_ exposure in January to April of 2020 compared to same month each year from 2016 to 2019.

	Model 1 ^(a)^	Model 2 ^(b)^	Model 3 ^(c)^	Model 4 ^(d)^
Reduction of PM_2.5_ by comparing 2016–2019 and 2020 (μg/m^3^)	−5.6 (−9.0, −2.3)	−4.1 (−7.2, −0.9)	−15.1 (−27.1, −3.2)	−11.2 (−14.3, −8.2)
Avoided cause-specific deaths				
Estimation using RRs from MCC study				
Non-accidental mortality	51.3 (44.6, 58.1)	37.6 (32.6, 42.5)	137.9 (119.8, 156)	102.5 (89.0, 115.9)
Cardiovascular disease mortality	9.6 (7.8, 11.5)	7 (5.7, 8.4)	25.7 (21.0, 30.8)	19.1 (15.6, 22.9)
Respiratory disease mortality	6.5 (4.6, 8.3)	4.7 (3.4, 6.1)	17.3 (12.5, 22.2)	12.9 (9.2, 16.5)
Estimation using RRs from Seoul City study				
Non-accidental mortality	25 (0.8, 49.8)	18.3 (0.6, 36.5)	67.2 (2.0, 133.9)	49.9 (1.5, 99.5)
Cardiovascular disease mortality	13.2 (2.1, 24.3)	9.7 (1.5, 17.8)	35.4 (5.6, 65.2)	26.3 (4.2, 48.5)
Respiratory disease mortality	15.3 (4.8, 25.8)	11.2 (3.5, 18.9)	41 (12.9, 68.6)	30.5 (9.6, 51.2)

^(a)^ Model 1 estimated by simple comparison of mean value, ^(b)^ Model 2 adjusting for daily average temperature, relative humidity, and wind speed, ^(c)^ Model 3 adjusting for daily average temperature, relative humidity, wind speed, years (in continuous variable), and months (as categorical variable), ^(d)^ Model 4 adjusting for daily average temperature, relative humidity, and wind speed and considered the effect of particulate filtering respirator.

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
