# Peer review of "Decrease in Ambient Fine Particulate Matter during COVID-19 Crisis and Corresponding Health Benefits in Seoul, Korea"

_ijerph, 2020, doi:10.3390/ijerph17155279_

Round 1

Reviewer 1 Report

As a reviewer I have the following remarks

  1. The paper is well written and presented.
  2. Line 85. “a reasonable assumption that social distancing and increased 85 telecommuting in Korea,” –for me it’s not clear how such restrictions reduce PM levels.
  3. Table 1. I suggest to simplify your names of the columns. For example: “Average daily PM2.5 (μg/m3), Mean (SD)”, just “PM2.5”. Description pleas add in the notes under the table, define SD.
  4. Table 2. Please consider the following. Year: 2016….2020. Also say Jan – January, etc.
  5. In general, in the paper there are many assumptions/extrapolations. The results strongly depend on them and models. Also an economic context – such type of the reduction is very expensive.Thank you.

Author Response

Thank you for the opportunity to revise our manuscript. We believe this revision made our manuscript much improved. We tried to respond to reviewers’ precious comments. We followed the International Journal of Environmental Research and Public Health’s submission guideline to prepare the revised manuscript.

Reviewer 2 Report

This is an innovative piece of research taking advantage of the unintended consequences of the COVID-19 pandemic of lockdown and less commuting on the health consequences of better air quality in Seoul, Korea.

What intrigued me, however, is why was Seoul chosen and not Busan? Perhaps this should be included. Currently, Seoul is only 39th on the World Air Quality and Pollution Index while Busan is 26th, thus much worse in terms of ambient fine particular matter.

To increase the relevance of the study applications to for instance four other Chinese cities like Bejing, Shanghai, Shenyang and Hangzhou could also be interesting. These four cities of a neighbouring country are all in the top 20 of most polluted cities in the world.

One would assume that because of lockdown measures, fewer factories were functioning which could have also contributed to cleaner air for 2020. Has this been factored in or at least mentioned in the study?

Specific comments:

1) Did you control for COVID-related deaths in Figure 2? In other words, the non-accidental, CVD and Respiratory mortality deaths, where people could also have died because of COVID in your estimations.

2) Line 259 mentions Table S2 and Figure S2 and Line 419 mentions www.mdpi where one can access these tables. However, I could not see any of these Supplementary Materials.

3) Line 497 indent  "Accessed 22 April 2020" to the right.

Author Response

(The authors gave the same response as above.)

Reviewer 3 Report

This study examined the effects of reduced air pollution during the first few months of 2020 on several population health indicators. The manuscript was well written, with results and figures presented clearly. The conclusions were well supported by the results produced. Here are some comments and suggestions to improve the manuscript:

Abstract

My suggestion to the authors is to insert a new first sentence on the main sources of air pollution and their exposure effect on population health. Then in a second new sentence, to explain how the COVID-19 situation influenced the level of activities that contributed to the main sources of air pollution in 2020. 

Introduction

Lines 83 to 83 - I think the authors mean "...suggest that measures to control COVID-19 may be the real reason behind the improved air quality"?

Lines 87 to 89 - Can the authors provide some evidence that the masks were "were effective in filtering"? If not, suggest to drop that phrase.

Lines 90 to 99 - These sound like Methods and would be better placed in the Methods section if not already there.

Methods

Lines 126 to 131 - As the lagged effects of PM2.5 exposure were not considered due to data unavailability, can the authors suggest how this might affect their study estimates? i.e. would they have under- or over-estimated the reductions in mortality due to lower PM2.5 exposure?

Line 195 - In the equation, I think the authors mean "Excess" instead of "Excessive"?

Line 201 - Change "morality" to "mortality". 

Lines 202 to 207 - This is the only major point that I am seeking clarification on. Can the authors give additional justifications for assuming that 0 µ/mgof PM2.5 is the minimum threshold effect? I would presume that previous time series studies on the health outcomes which the dose-response relationships are assumed in this study, did not have 0 µ/mg3 observations. Would it be more reasonable to assume that the minimum threshold effect would be the minimum levels observed in those studies?

Author Response

Thank you for the opportunity to revise our manuscript. We believe this revision made our manuscript much improved. We tried to respond to reviewers’ precious comments. We followed the International Journal of Environmental Research and Public Health’s submission guideline to prepare the revised manuscript. Please see the attachment.

Round 2

Reviewer 3 Report

The authors have addressed my comments satisfactorily and I would recommend that this manuscript be accepted for publication.